# High-Speed Railway Network, City Heterogeneity, and City Innovation

**Kunlun Zhao * and Wenxing Li**

School of Economics and Management, Beijing Jiaotong University, Beijing 100044, China; wxli@163.com
* Correspondence: 19113024@bjtu.edu.cn

**Abstract:** The emergence of the time–space contraction effect from the high-speed railway (HSR) network in China has been beneficial in breaking down regional divisions, thus facilitating the circulation of resources and optimizing resource distribution and production efficiency. However, research has not adequately addressed the city disparities of the HSR network and their effects on city innovation. Through the heterogeneity perspective of 'New' new economic geography, this study employs the 2008–2019 panel data at the city level in China and builds a spatial Durbin model based on continuous spatial difference in differences to investigate the mechanism of the HSR network on city innovation and to analyze its agglomeration and diffusion effect of innovative factors under different city sizes and spatial perspectives. This study revealed that the HSR network could significantly increase the innovation of local cities and neighboring cities, yet there is a certain threshold of city size that affects city innovation. Large cities covered by HSRs can take advantage of gathering talent, financial capital, and industry from nearby regions, thus constructing a new spatial pattern of innovative development. This study also found that the innovation accelerative effect gradually decreases as the distance from the city covered by HSRs increases and completely disappears at the distance of 400 km. Therefore, it is necessary to optimize the HSR network and increase the mobility and agglomeration of innovative elements between cities, thus deepening the collaboration between cities through differentiated strategies. This will enhance the spatial spillover effect of innovation, thus ultimately achieving a balanced spatial pattern of city innovation.

**Keywords:** HSR networks; city heterogeneity; city size; city innovation

## 1. Introduction

In 2016, China introduced the "National Innovation Driven Development Strategy Outline", making innovation a popular subject in Chinese economic research. Economically speaking, a combination of factors such as talent, capital, and technology is a major component in fostering innovation. According to the endogenous growth theory, regional innovation growth is mainly due to R&D investment and knowledge spillovers [1]. However, the mobility of resources, labor, and capital is restricted by distance, making it difficult to form an appropriate spatial distribution structure. Optimizing the spatial arrangement of innovative resources and increasing city innovation efficiency is essential for building an innovative nation and achieving innovation-driven development in China.

HSRs have been identified as one of the most revolutionary and innovative forms of transportation in the 21st century. They have been shown to reduce the costs of knowledge spillover by shortening spatial and temporal distances, as well as accelerating the flow of innovative elements such as talent and capital between cities [2,3]. Additionally, they have been found to influence the spatial distribution of economic factors, thus reshaping the regional spatial development structure. Furthermore, HSRs have been observed to reduce the cost of trade between cities [4,5]. The New Economic Geography theory suggests that when trade costs are reduced, production factors will be relocated in order to increase

actual profits, and the relocation of innovative elements across regions will cause a spatial differentiation of city creativity [6].

HSRs can be beneficial for economic growth and the aggregation of economic activities. The network's characteristics enable the transfer of production factors from one area to another, and cities that are close to each other are more likely to take advantage of the resources of the central city, resulting in a positive spatial spillover effect [7,8]. However, in regions with a well-developed HSR network, the economic agglomeration effect of the central city can also impede the economic development of the surrounding cities, resulting in a negative spatial spillover effect [9]. This can be detrimental to poorer areas, as they cannot benefit from the increased investment in nearby areas The rationale explanation for coming to a conflicting conclusion is that various cities have diversity at different stages, and blurring different spatial patterns will cause a spatial mismatch in the innovative developments of cities. Exploring the innovative difference and its mechanism between different city sizes is the most urgent problem that needs to be solved for city innovation from a global perspective.

This article proposes that in order to reduce uncertainty in the innovative developments of cities, mapping the stages of city development to changes in the spatial structure should be conducted, which can reveal the relationship between the dynamic changes caused by the construction of the HSR network and the innovation output of different cities. Thus, it is necessary to consider how the implementation of the HSR network changes the spatial pattern of city innovation and boosts the spatial allocation of innovative resources between cities. Ignoring this would hinder the efficient allocation of innovative resources by the government and enterprises, as well as the optimization and balanced growth of innovation patterns between cities.

Industrial transfer is typically arranged geographically based on the cost-minimizing principle, meaning even smaller-sized cities can take part in specific industrial transfers due to their lower production and living costs [10]. However, innovation activities have a higher skill threshold, and the time cost for talents is more significant, so the skill threshold, due to different city sizes, can be a large limitation on innovation [11]. Numerous facts point to the fact that global innovation resources are increasingly concentrated in cities with a great deal of talent and technological capital. These cities not only demonstrate individual innovation strength but also demonstrate collective innovation abilities However, in cities that have been completely covered by HSRs, smaller cities usually cannot take advantage of the innovation opportunity that the HSR brings, which accounts for these disparities between different city sizes.

In the last few years, research on city agglomeration has revealed that the notion of homogeneity among enterprises is flawed and that disparities in enterprise efficiency have a major effect on city innovative efficiency. Researches [12–14] have proposed more realistic hypotheses in this regard. Agglomeration economies can certainly increase enterprise productivity, but the 'New' new economic geography theory also indicates that the high cost and competitive environment of large cities can act as a filter for enterprise production efficiency. As the market size grows, the competition among firms intensifies, leading to a decrease in the critical marginal production cost that determines which firms can survive. Firms that fail to reach this cost threshold will not be able to survive in this market. This explains why global innovation resources are increasingly concentrated in large cities, but it still does not clearly account for the mechanism that HSRs use to promote agglomeration and diffusion between different innovation elements. There are few studies that focus on heterogenous city innovation from the spatial perspective. The traditional viewpoint of knowledge spillover and agglomeration does not consider the city heterogeneity and spatial perspective simultaneously, so it does not exactly capture the direction of factor agglomeration and diffusion by using the traditional analysis method and perspective. However, heterogeneity is the core of city innovation for determining the direction of factor agglomeration and diffusion in the real world.

This investigation has established a robust theoretical basis regarding the spillover effect of transportation infrastructure and city innovation. Nevertheless, most of the current studies are based on the homogeneity hypothesis of the new economic geography theory and neglect the threshold effect of different city sizes on the selection of innovative elements. The spatio-temporal compression effect of HSR networks significantly affects the circulation of innovative elements and the spatial arrangement of innovation capacity, which have yet to be fully understood. This article examines the impact of HSR networks on city innovation by utilizing the continuous spatial difference-in-differences (CSDIDs) method. This method is an improvement on the DIDs, which only uses the dummy variable for whether the HSR is open or not in previous studies. The CSDIDs method takes into account the annual construction process of HSR networks and the gradual state between cities, allowing for the dynamic process of HSR network expansion to be taken into account. This method also establishes a connection between the connectivity of city networks in different spatial structure changes and innovation outputs, thus providing a better understanding of the structural effects of HSR networks on city innovation.

The marginal contributions of this paper are as follows: First, traditional studies on the impact of HSRs on city innovation based on dummy variables ignore the dynamic change process of HSR network expansions. This paper identifies the impact of HSR networks on city innovation and their spatial spillover effect through the CSDID method based on the HSR network index instead of dummy variables for whether the city has been covered by the HSR. Second, most of the existing studies studied the impact of HSRs on city innovation under the assumption of homogeneity, and they ignore the city heterogeneity regarding the differences in receiving and transforming knowledge under different city sizes. This paper examines the heterogeneity effect of HSR networks on city innovation under the moderating effect of city size, which demonstrates the agglomeration and diffusion difference between different city sizes. Third, previous studies have demonstrated the mechanism of city innovation by factor agglomeration and knowledge spillover, but these still do not know the path of the innovative elements of agglomeration and diffusion in the HSR network and the criteria for determining the direction of factor agglomeration and diffusion. This paper adopts CSDIDs to explore its spatial mechanism and the agglomeration and diffusion path of innovation elements, providing a theoretical basis for spatial allocation of innovation resource under different city sizes. Finally, this study further deepens the theory of geographic economics by proposing the select effect of city size when HSR networks facilitate the element flow, which not only focuses on the agglomeration and diffusion effect of absolutely large and small cities, but it should also carry out in-depth promotions in combination with China's "Metropolitan Circle" to form an efficient and comprehensive network structure of city innovation based on the ordered city size and the city's functions and roles.

## 2. Data and Methods

### 2.1. Analysis Procedure

It must be noticed that most studies have generally utilized dummy variables to indicate the presence of HSRs [15,16]. However, this method does not take into account the heterogeneity of proximity between cities in terms of the level of HSR network connectivity [17], as a larger transportation network connectivity is more effective in advancing factor flows and thus exercising a greater influence. This article proposes an HSR network index based on the number of connections between cities through HSRs. Therefore, based on the concept of continuous DIDs, this paper adopts CSDIDs to analyze the impact of an HSR network on city innovation [18]. Compared with binary SDIDs, CSDIDs do not change the original attribute characteristics of DIDs but can show more abundant data characteristics and help to avoid the potential bias in the process of subjective settings of the experimental group and the control group.

This study investigated the relationship between HSR networks and city innovation by utilizing four models. These models included the general OLS model based on local

effects and three models with special parameters: the Spatial Durbin Model (SDM), the Spatial Autoregressive Model (SAR), and the Spatial Error Model (SEM). The SDM was the most comprehensive model. The SEM focused on the spatial correlation of the error term. Equations (1) and (2) demonstrate the CSDIDs model.

$$y_{it} = \alpha + \rho \sum_{j=1}^{n} w_{ij} y_{jt} + \beta x_{it} + \theta \sum_{j=1}^{n} w_{ij} x_{it} + \mu_i + \gamma_t + u_{it} \tag{1}$$

$$u_{it} = \lambda \sum_{j=1}^{n} w_{ij} u_{jt} + \varepsilon_{it} \tag{2}$$

The innovation output level of city $i$ in year $t$ is represented by $y_{ij}$. $x_{it}$ is a vector of variables at the city level, comprising the HSR network index, the city's size of population, the city's fixed asset investment, and the GDP at the city level. $w_{ij}$ symbolizes the standardized weight matrix at the city level. $\rho$ and $\theta$ are two spatial lag parameters, capturing the strength of the spatial dependence of $y$ and $x$ at the two spatial scales. $\lambda$ is the autocorrelation parameter for the spatial error term, which is used to measure the intensity of the unobserved factors' spatial dependence. $\mu_i$ is the city's fixed effect setting, $\gamma_t$ is the year's fixed effect setting, and robust standard errors are clustered to the provincial level. $u_{it}$ is a random factor that causes errors.

If $\rho$, $\theta$, and $\lambda$ are all equal to 0, Equation (1) estimates the local direct effect by a continuous DID estimation. If $\theta$ and $\lambda$ equal to 0, Equation (1) illustrates the spatial autocorrelation of SAR estimation. When $\rho$ and $\theta$ in Equation (1) equal to 0, Equations (1) and (2) reflect the spatial dependence of unobserved factors by the SEM model.

Additionally, this article investigates the radiation and siphon effects of HSR networks on city innovation based on the threshold effect of city size by comparing different subgroups, as well as the attenuation effect of HSR networks on city innovation based on different weight matrices constructed by the distance threshold, and this is used to determine the radiation and siphon effect boundaries of city innovation effects.

### 2.2. Variable Construction

### 2.2.1. The Level of City Innovation

Innovation usually includes innovation input and innovation output, while, due to the risks associated with such innovation activities, many of the inputs may not result in any practical output. Moreover, the number of patent applications may not accurately reflect its level of innovation output. Patent authorization can accurately reflect the level of city innovation, but obtaining a patent requires testing, annual fees, and is susceptible to political interference, making it an unpredictable process. Employing the variable of authorized patents to measure the city's innovation level has many drawbacks, such as the incapacity to show their true social and economic value and the difficulty of horizontally comparing patents from different industries. In this paper, the innovation index of 286 cities from 2008 to 2019, calculated by the Construction and Framework of China's Regional Innovation Index, is used as the variable of the city's innovation level. This index estimates the patent value through a patent renewal model and aggregates it to the city level, thus providing an effective solution to the issue of patent quality heterogeneity and obtaining the city's innovation index. This paper opts for the innovation patent index as the proxy for city innovation outputs and uses the number of patent authorizations as a robustness test. To account for the time lag of HSRs in the production and innovation processes and to avoid any reverse causal effects, the econometric model uses a one-phase lag of patent variables. The Moran index and scatter plot in Figure 1 show that for the innovation index and authored patents, there exists a significant positive correlation at the spatial level, which provides the basis of analyzing city innovation from the spatial perspective.

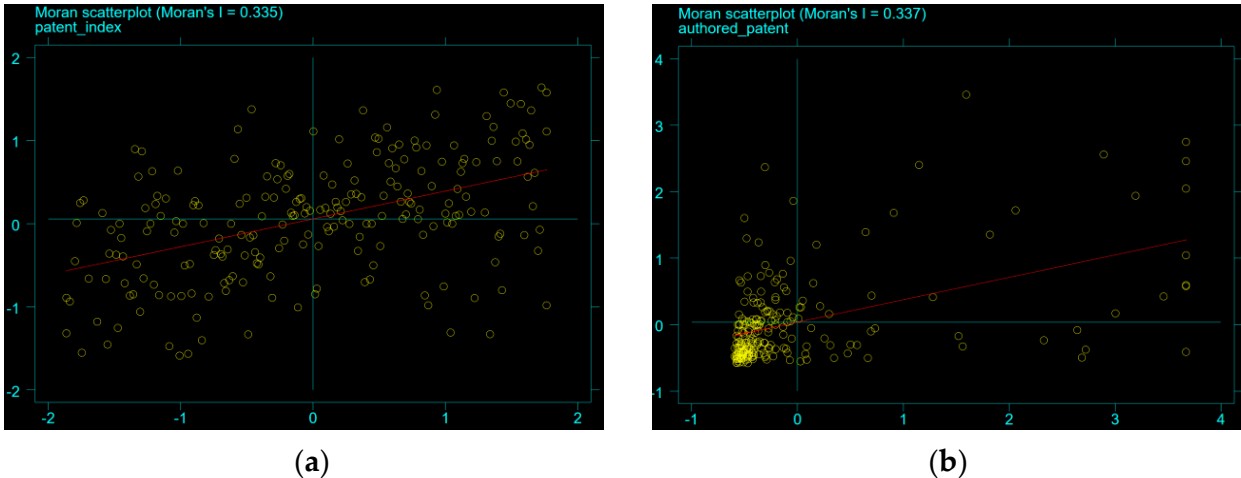

(**a**)            (**b**)

**Figure 1.** Scatter plot of patent index and authored patents based on spatial contiguity matrix in 2019. (**a**) Innovation patent index; (**b**) Authored patents.

2.2.2. HSR Network Index

HSR networks can be analyzed by using a complex network method, which is helpful to better understand the structure and changes of HSR networks [19]. When using complex networks, the topology properties of networks can be divided into node attributes and structure attributes. The value of a node in a network depends on its location in the network. The importance of a node is usually measured by its centrality index. There are different kinds of network centralities, such as the degree centrality (DC), the Closeness centrality (CC), the Betweenness centrality (BC), and the eigenvector centrality (EC).

DC refers to the number of other cities connected to the node city. The more the number of connections, the more convenient the site is to travel.

$$DC_i = \sum_{i \neq j}^{n} x_{ij} \tag{3}$$

where $x_{ij}$ indicates whether cities $i$ and $j$ are directly connected by the HSR, and $n$ represents the number of cities.

CC can be judged by the spatial position of the node city, which is the sum of the distance between a node and all other nodes. The smaller the sum is, the shorter the path from this node to all other nodes is. In other words, the closer this node is to all other nodes, and the node city is closer to the center of the network.

$$CC_i = \frac{n-1}{\sum_{i \neq j}^{n} d_{ij}} \tag{4}$$

where $d_{ij}$ is the distance from city $i$ to city $j$.

BC is a more complex concept, yet it is of greater practical importance, as it reflects the load on nodes in the network, that is, how many routes pass through the station. Therefore, the more paths that pass through a node, the higher its BC.

$$BC_i = \sum_{i \neq j}^{n} \frac{\sigma_{ij}^m}{\sigma_{ij}} \tag{5}$$

The number of shortest paths connecting cities $i$ and $j$ is denoted by $\sigma_{ij}$, while the number of the shortest paths that must pass through node $m$ in the shortest path from node $i$ to node $j$ is represented by $\sigma_{ij}^m$.

EC is a measure of the influence of vertices in a network, where the centrality of a node is determined by the centrality of its adjacent nodes. Nodes with higher scores of

adjacent nodes will have higher scores than those with lower scores. Therefore, all nodes will be assigned a score based on this principle; a higher score of the eigenvector indicates that the node is connected to many nodes with higher scores. The more important the nodes are connected to, the more important the city *i* becomes. An eigenvector score is used to measure the degree of connections between each vertex, and this score is then used to reassign the summation value of adjacent nodes to each point, thus increasing the degree centrality.

$$EC_i = c \sum_{j=1}^{n} a_{ij} DC_j \qquad (6)$$

$c \neq 0$ is a constant, and $A = a_{ij}$ is a matrix that is determined by whether the city pairs are directly connected.

The construction of an HSR network index requires changing the connectivity information between stations into a network form. In this paper, cities connected by HSR lines are selected as network nodes, and all prefecture-level cities are included. Regardless of whether these cities have been connected to the HSR network or not, all types of bullet trains (classified as C, D, and G in China) are considered as HSRs. Therefore, a matrix of 286 cities is constructed in this paper. If these city pairs are directly connected, each indicator of a city is assigned a value of 1; otherwise, it is assigned a value of 0. If a city has multiple HSR stations, no matter how many HSR stations the city has, this paper identifies the city as only one node. For example, the Beijing Railway Station, the Beijing West Railway Station, the Beijing South Railway Station, and the Beijing North Railway Station in Beijing are merged into one node in the network.

The data source of the HSR city pairs is the website of China Railway Ticketing (www.12306.cn), which contains information of all the HSRs in operation, including the opening time, routes, and stations (excluding Hong Kong, Macao, and Taiwan). Finally, the Gephi-0.9.6 software was used to calculate these indexes.

The map in 2019, showed in Figure 2, revealed that the eastern region and the coastal region in China have a higher HSR network index, while these areas usually have a higher innovation capacity, so we will use the spatial model to explore the casual effect between the HSR network and the city's innovation index.

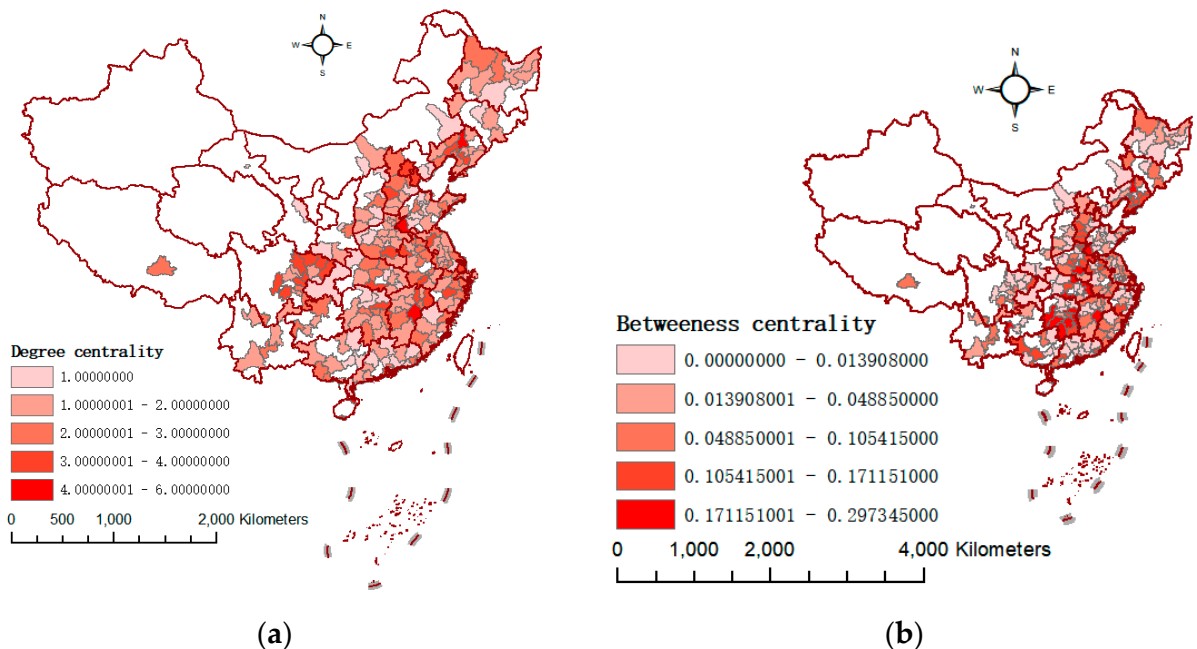

**(a)** **(b)**

**Figure 2.** *Cont.*

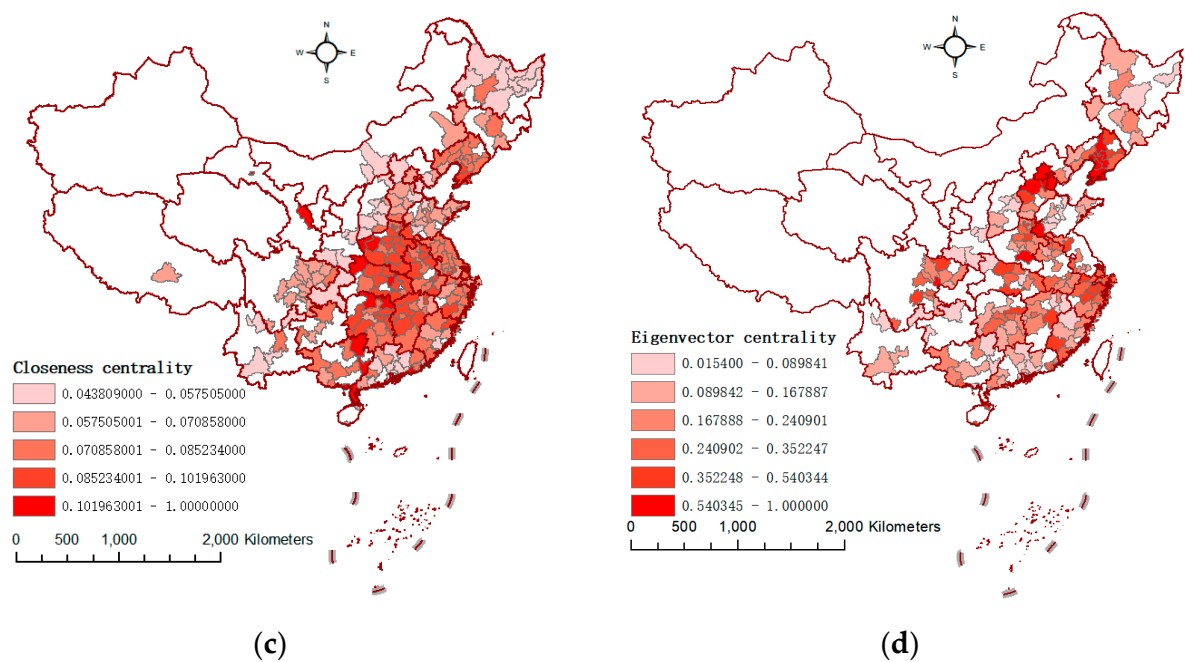

**Figure 2.** The distribution of HSR network centrality index in 2019. (**a**) Degree centrality in 2019; (**b**) Betweenness centrality in 2019; (**c**) Closeness centrality in 2019; (**d**) Eigenvector centrality in 2019.

2.2.3. Covariates

Covariates include the population size, the city's fixed asset investment, and the GDP. Considering the technological gap between China's coastal areas and inland areas, the closer the location is to coastal port cities, the better the level of economic development. Therefore, the distance between the local city and the nearest coastal city is taken as the proxy variable of the local economic development level and technological difference instead of the GDP as the measurement directly, so as to eliminate the endogenous problems that may exist between the direct use of the GDP variable and the city's innovation level. In the case of outliers in some indicators, the tail reduction of 0.5% is applied to all continuous variables.

Descriptive statistics of the variables used in this article are presented in Table 1.

**Table 1.** Descriptive statistics.

| Variables | Mean | SD | Variables | Mean | SD |
|---|---|---|---|---|---|
| Authorized patent | 3820 | 8448 | Appearance patent index | 51.85 | 28.20 |
| Innovation index | 52.07 | 28.02 | Authorized trademark index | 52.07 | 28.01 |
| Index of per capita | 51.32 | 28.46 | Degree centrality | 0.01 | 0.03 |
| Index of unit area | 51.68 | 28.07 | Closeness centrality | 0.09 | 0.19 |
| Index of new enterprises | 51.95 | 27.99 | Betweenness centrality | 0.016 | 0.04 |
| Index of foreign investment | 51.66 | 28.32 | Eigenvector centrality | 0.10 | 0.17 |
| VCPE investment index | 51.52 | 28.61 | Investment in fixed assets | 1390 | 1457 |
| Index of invention patent | 52.07 | 28.08 | Distance to port city | 613.0 | 393.6 |
| Utility patent index | 52.13 | 28.00 | Registered population | 434.6 | 266.1 |

## 3. Results

### 3.1. Model Suitability Test

To assess the efficacy of the spatial model settings presented in this paper, a series of applicability tests were conducted on the spatial model in Table 2. The SDM is employed in this article, and the Hausman test is conducted to decide between fixed and random effects. The Hausman test results indicate a significant estimation, implying that the fixed effects model is suitable. Subsequently, Wald and LR tests were conducted on the SDM to determine if it would be weakened into the SAR or SEM. The LR-Error, Wald-Error, LR-Lag, and Wald-Lag test values are all significant at the 1% level, implying that the SDM will

not deteriorate into the SAR and SEM, so the SDM should be taken into consideration [20]. Therefore, this study uses the SDM with both time and individual fixed effects for a regression analysis.

**Table 2.** Model suitability test.

|  |  | Adverse Distance Matrix | | Adjacency Matrix | |
| --- | --- | --- | --- | --- | --- |
| **Centrality** | **Method** | **Sample Size** | **$p$** | **Sample Size** | **$p$** |
| Degree | LR-Lag | 19.22 | 0.000 | 19.22 | 0.000 |
| centrality | LR-Error | 16.31 | 0.001 | 16.31 | 0.001 |

*3.2. Baseline Regression*

This article uses CSDIDs to investigate the influence of HSR networks on city innovation. It needs to be noticed that all cities have the same changing trend as before the opening of the HSR, and a parallel trend test was conducted between city innovation and HSRs in Figure 3. A dummy variable was used to distinguish between the control group and the treated group. The results of the parallel trend test show that there is not a significant difference in all pre-treatment coefficients of HSRs on city innovation. However, after the city was covered by the HSR, there was a significant difference in the one-phase lag, indicating that the impact of HSRs on city innovation meets the parallel trend assumption.

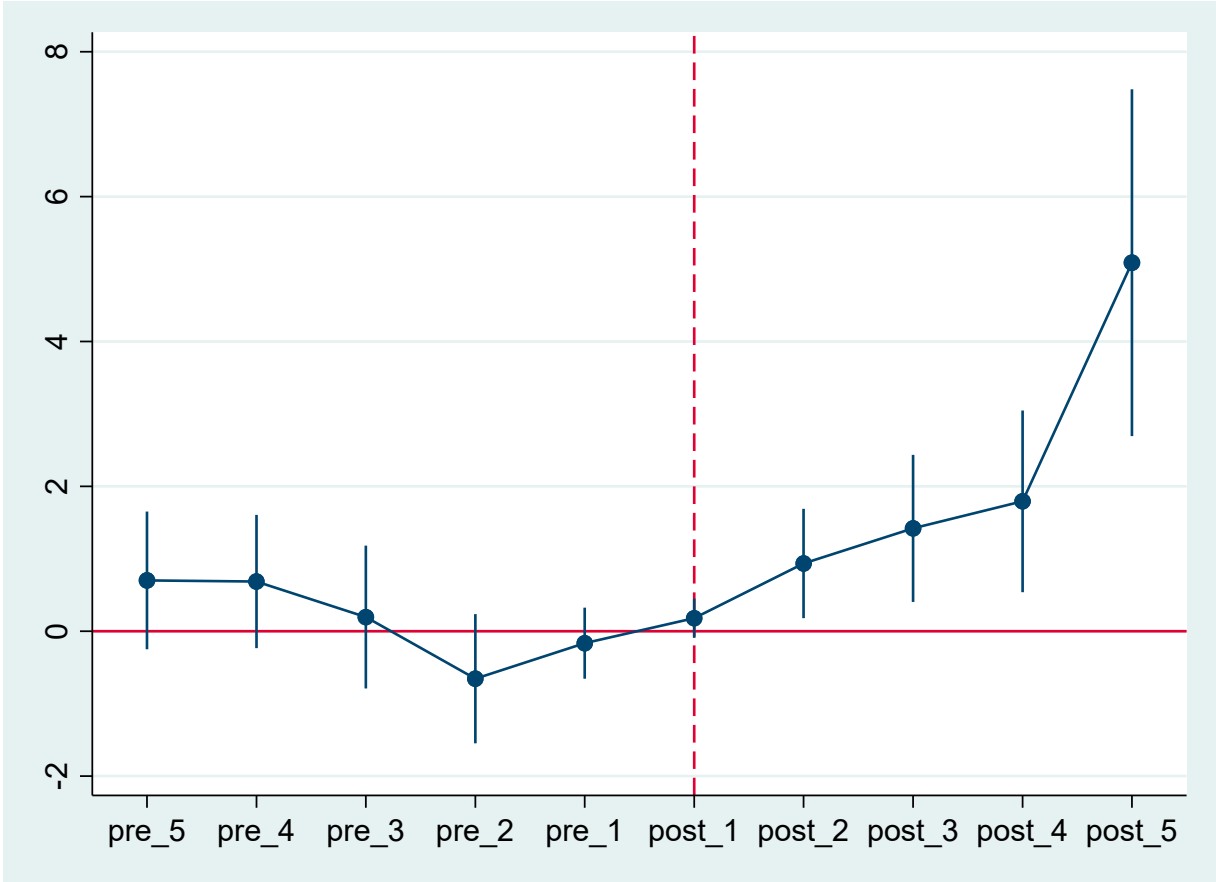

**Figure 3.** Parallel trend test of city innovation.

Table 3 presents the DIDs' estimation of city innovation based on the degree centrality network index of HSR. The first column of Table 3 shows the continuous DIDs' estimation without considering spatial effects, which reveals that the HSR network has a significant positive effect on city innovation. Specifically, for every unit increase in the centrality index



of the HSR network, the city innovation index increases by 26.31 points. To address the spatial spillover effect of city innovation and the network attributes of HSRs, the CSDIDs method is used in the second column of Table 3, which simultaneously considers the spatial autocorrelation and spatial spillover of city innovation. The results demonstrate that the coefficient of $W$*Degree is significant and positive at the 1% level, indicating that HSR networks can not only promote innovation in the local city but also have a positive effect on the innovation of the surrounding cities. Moreover, it was observed that there is an aggregation of innovation activities in adjacent cities, with cities exhibiting notable innovation activities congregating together, thereby furthering city innovation through the population's communication and knowledge spillover under the HSR network.

**Table 3.** The effects of HSR networks on city innovation.

|  | OLS | SDM | SAR | SEM | SDM |
|---|---|---|---|---|---|
| Degree | 26.31 *** | 4.298 *** | 22.44 *** | 20.42 *** | 3.565 *** |
|  | (2.48) | (1.89) | (6.43) | (5.22) | (1.42) |
| $\theta$ ($W$*Degree) |  | 54.79 *** |  | 0.023 *** |  |
|  |  | (6.02) |  | (0.011) |  |
| $\theta$ ($W_i$*Degree) |  |  |  |  | 183.5 *** |
|  |  |  |  |  | (13.2) |
| $\rho$ (Innovation Index) |  | 0.181 *** | 0.203 *** |  | 0.394 ** |
|  |  | (0.040) | (0.049) |  | (0.155) |
| $\lambda$ |  |  |  | 0.196 |  |
|  |  |  |  | (0.248) |  |
| Control variable | √ | √ | √ | √ | √ |
| Fixed time | √ | √ | √ | √ | √ |
| Fixed city | √ | √ | √ | √ | √ |
| Obs | 2508 | 2508 | 2508 | 2508 | 2508 |
| $R^2$ | 0.517 | 0.091 | 0.116 | 0.062 | 0.103 |

Note: Standard errors are reported in the parentheses and clustered at the provincial level; *** $p < 0.01$, ** $p < 0.05$, * $p < 0.1$.

The SDM model revealed that the estimated result of the HSR network on city innovation was significantly lower than the estimated results of the OLS, SAR, and SEM, suggesting that omitting the spatial effects of city innovation could lead to an overestimation of the influence of the HSR network on city innovation. This could be because city innovation is diffused to neighboring cities through knowledge spillover, thus reducing the effect of HSRs on local city innovation.

This study further investigates the impact of the HSR network on city innovation and its spatial effect by utilizing the geographical inverse distance matrix. The results from the fifth column of Table 3 demonstrate that the HSR network index still has a significant effect on the promotion of city innovation, and there is a significant spatial spillover effect on the innovation activities of neighboring cities. Moreover, the innovative activities of neighboring cities also exhibit spatial agglomeration by using the geographical inverse distance matrix.

The findings of Table 2 demonstrate that the introduction of HSR networks has a positive impact on innovation capabilities in the local area and neighboring cities, as well as on the synchronization of innovation between cities. Generally, the diffusion of innovative knowledge is hampered by geographical barriers, but the HSR network significantly reduces these barriers, thus facilitating knowledge diffusion. The connection of the HSR network has weakened the influence of spatial barriers on city innovation. Moreover, the results in Table 3, based on both the spatial weights of contiguity and the geographical inverse distance, indicate that HSR networks significantly improve city innovation.

### 3.3. Robustness Test

The importance of nodes in the structure of HSR networks can be expressed in various forms. DC represents the travel convenience of a city covered by an HSR, while BC represents the importance of the spatial location covered by an HSR, which is similar to the concept of accessibility. CC reflects the potential load of nodes in the HSR network. That is, the number of travel paths undertaken by the node city, and EC depends on the connection with the node city with a higher score. This article further uses DC, CC, and EC to explore the impact of HSR networks on city innovation. The results in Table 4 demonstrate that the HSR network has a positive effect on the innovation activities of local cities and their surrounding cities and that the spatial aggregation of city innovation is still present, thus confirming the efficacy of the HSR network on city innovation.

**Table 4.** The influence of different HSR network indexes on city innovation.

| | Innovation Index | | | |
|---|---|---|---|---|
| HSR | 0.869 *** | | | |
| | (0.020) | | | |
| $\theta$ (W*HSR) | 0.083 *** | | | |
| | (0.028) | | | |
| Betweenness centrality | | 2.639 *** | | |
| | | (0.803) | | |
| $\theta$ (W*Betweenness centrality) | | 10.79 *** | | |
| | | (1.78) | | |
| Closeness centrality | | | 0.178 ** | |
| | | | (0.071) | |
| $\theta$ (W*Closeness centrality) | | | 1.624 * | |
| | | | (0.925) | |
| Eigenvector centrality | | | | 0.585 *** |
| | | | | (0.058) |
| $\theta$ (W*Eigenvector centrality) | | | | 5.167 ** |
| | | | | (2.622) |
| $\rho$ (Innovation Index) | 0.188 *** | 0.188 *** | 0.188 *** | 0.185 *** |
| | (0.045) | (0.045) | (0.046) | (0.043) |
| Control variable | √ | √ | √ | √ |
| Fixed time | √ | √ | √ | √ |
| Fixed city | √ | √ | √ | √ |
| Obs | 2508 | 2508 | 2508 | 2508 |
| $R^2$ | 0.074 | 0.083 | 0.078 | 0.095 |

Note: Standard errors are reported in the parentheses and clustered at the provincial level; *** $p < 0.01$, ** $p < 0.05$, * $p < 0.1$.

City innovation should be a comprehensive result in multiple dimensions, which cannot be simply quantified by the number of authorized patents and applied patents. In this paper, the number of authorized patents and the categories of authorized patents are selected as the proxy variable of the innovation output for the robustness test. The results of Table 5 show that the HSR network not only increases the overall innovation level of cities but also boosts the number of invention, practical, and appearance patents, as well as the trademark authorizations. Additionally, it is evident that HSR networks have a positive spatial correlation of authorized patents between cities, further confirming the result that HSR networks promote city innovation.

### 3.4. Endogenous Problems

This article suggests that improving HSR networks can promote city innovation, while there may be endogeneity issues between the construction of the HSR network and the degree of city innovation, which could lead to a bias in the estimation results. This article takes into account the potential endogeneity selection problem of HSR openings by using the instrumental variable (IV) method. Even though the construction of HSRs is assumed

to be exogenous to cities along the HSR line, cities that are covered by HSRs usually have high levels of economic development and innovation potential. Therefore, the estimation results could be biased. Generally, the steeper the terrain of a city, the more challenging it is to construct an HSR. As the geographical factor that does not directly influence city innovation, this study uses the geographic slope to create IVs in order to address potential endogeneity issues. Furthermore, the current railway lines are correlated with historical railway lines, and it is difficult to affect current city innovation through channels other than HSR connections [8]. Therefore, this paper further utilizes historical railway data from 1961 to create IVs to provide a robustness test for the endogeneity of the empirical estimation.

**Table 5.** Influences of HSR networks on the different dimensions of city innovation.

|  | **Authorized Patent** | **Standardized City Innovation Index Score** | | | |
|---|---|---|---|---|---|
|  |  | **Invention Patent** | **Utility Patent** | **Appearance Patent** | **Trademark Authorization** |
| Degree | 4.298 ** | 3.383 ** | 1.700 *** | 7.388 *** | 3.556 *** |
|  | (1.89) | (1.40) | (0.60) | (2.50) | (1.01) |
| $\theta$ | 54.79 *** | 71.34 *** | 84.00 *** | 58.78 | 33.22 * |
|  | (16.02) | (2.63) | (25.65) | (39.89) | (17.79) |
| $\rho$ | 0.181 *** | 0.231 *** | 0.287 *** | 0.204 *** | 0.155 *** |
|  | (0.040) | (0.073) | (0.053) | (0.048) | (0.032) |
| Control variable | $\checkmark$ | $\checkmark$ | $\checkmark$ | $\checkmark$ | $\checkmark$ |
| Fixed time | $\checkmark$ | $\checkmark$ | $\checkmark$ | $\checkmark$ | $\checkmark$ |
| Fixed city | $\checkmark$ | $\checkmark$ | $\checkmark$ | $\checkmark$ | $\checkmark$ |
| Obs | 2508 | 2508 | 2508 | 2508 | 2508 |
| $R^2$ | 0.091 | 0.082 | 0.122 | 0.116 | 0.098 |

Note: Standard errors are reported in the parentheses and clustered at the provincial level; *** $p < 0.01$, ** $p < 0.05$, * $p < 0.1$.

Table 6 presents the results of the IV analysis. The Hausman test revealed a large discrepancy between the OLS and IV estimates. The F-test statistic was higher than the benchmark of 10, while the KP test indicated that the IV estimation based on the geographical slope and historical railway construction was not impaired by weak instruments. The positive and significant IV estimation suggests that improving the HSR network may facilitate city innovation.

**Table 6.** IV estimates the impact of HSRs on the city innovation index.

|  | **Innovation Index** | |
|---|---|---|
| Degree | 87.94 *** | 5.74 *** |
|  | (7.533) | (0.541) |
| First stage | | |
| Slope | 0.003 *** | |
|  | (0.001) | |
| Rail 1961 | | 0.005 * |
|  | | (0.003) |
| F-statistic | 27.87 | 27.45 |
|  | (0.000) | (0.000) |
| KP test statistic | 27.65 | 16.36 |
|  | (0.000) | (0.000) |
| Control variable | $\checkmark$ | $\checkmark$ |
| Fixed time | $\checkmark$ | $\checkmark$ |
| Fixed city | $\checkmark$ | $\checkmark$ |
| Obs | 2508 | 2508 |
| $R^2$ | 0.497 | 0.235 |

Note: Standard errors are reported in the parentheses and clustered at the provincial level; *** $p < 0.01$, * $p < 0.1$.

*3.5. Spatial Attenuation*

The neighboring innovation effect of HSR networks is in line with the local effect, that is, it can help to improve the innovation level of neighboring cities. This paper explores the optimal radius of this effect and whether it varies depending on the size of the city. To do this, a distance threshold is established based on the geographical weight matrix, which is constructed with 50 km as the gradient threshold in turn. The spatial attenuation effect of HSR networks on city innovation is then examined, and the difference of the attenuation effect is tested by different city sizes.

Let the minimum distance between two cities be $d_{min}$ and the maximum distance be $d_{max}$, and use $d_{min}$ as the initial value, and increase $\tau km$ each time. According to $W_{it} = 1/d_{it}$, when $d_{it} < d$, the matrix element is set to 0, and the different distance threshold $d_{it} = d_{min}, d_{min} + \tau, d_{min} + 2\tau \ldots, d_{max}$.

The gradient weight matrix is set up as follows.

$$W_{ij}^d = \sum \begin{matrix} \frac{1}{d_{ij}}, & d_{ij} \geq d \\ 0, & d_{ij} \leq d \end{matrix} \qquad (7)$$

In Formula (7), $W_{ij}^d$ represents the spatial inverse distance matrix with the distance threshold, and d is the critical distance threshold. Since there are few cities within 100 km, a distance matrix of 50–750 km is obtained by taking 100 km as the initial value and 50 km as the step gradient.

Table 6 suggests that the HSR network has a negative influence on the innovation coefficient of cities within 100 km, while the coefficient of cities between 100–400 km is positive and statistically significant. As the distance threshold increases, the spatial spillover effect of the HSR network diminishes. For samples further than 100 km from the central city, the estimated coefficient decreases with the increase of distance, displaying a distance attenuation effect that is no longer significant in cities beyond 400 km. This research suggests that the construction of an HSR network has a siphoning effect on nearby cities, resulting in a negative spatial spillover on their innovation. However, as the HSR network further expands, its positive innovation spatial spillover effect decreases and eventually dissipates at a threshold of 400 km. This indicates that the optimal distance for the HSR network to have a positive effect on the innovation of neighboring cities is 400 km. The innovation spatial spillover effect of the HSR network shows a distance attenuation characteristic. An HSR network strengthens the economic ties between cities, but this has a negative impact on the innovation of cities within 100 km. As the centripetal and centrifugal forces transform, the radiation effect of the HSR network on city innovation begins to dominate, promoting the innovative behavior of surrounding cities, while this effect weakens with an increasing distance and dissipates at 400 km.

From the view of innovation demand, the influence of an HSR network on city innovation is related to the size of the city. Due to enterprises, human capital and cities with different sizes have different acceptances and transformations of innovation factors, and these differences will further affect the development of city innovation, that is, the role of an HSR on city innovation has a threshold effect of city size. Bigger cities are able to accept and transform more innovative elements, making them more attractive to innovation, whereas smaller cities have limited economic activity and their demand for innovative elements is lower, even affected by the "siphon effect" of larger cities. This article further divides cities into two groups based on their GDP ranking in the base period and examines the innovation spillover effect of the HSR network on neighboring cities in each size. In columns (2) and (3) of Table 7, although there is still a siphon effect on the neighboring cities for larger cities, its positive innovation spillover effect has a greater spatial attenuation distance and is no longer significant when the threshold is up to 500 km. On the other hand, smaller cities do not experience any spatial spillover effect of HSR networks on innovation in surrounding cities. This is due to the heterogeneous innovation capabilities among

different city sizes, which results in a threshold effect of the innovation spatial spillover of HSR networks on neighboring cities based on city size.

**Table 7.** Spatial attenuation effect boundary of HSR network.

|  | Total | Large City Size | Small City Size |
|---|---|---|---|
| >100 km | −4.08 ** | −7.32 ** | 5.46 |
| >150 km | 74.23 ** | 91.79 *** | 11.45 |
| >200 km | 65.38 *** | 76.43 ** | 8.54 |
| >250 km | 48.54 *** | 57.86 | 6.68 |
| >300 km | 32.98 ** | 48.42 ** | 8.84 |
| >350 km | 18.96 * | 38.27 * | 9.23 |
| >400 km | 5.88 * | 15.18 *** | 5.34 |
| >450 km | 4.87 | 16.47 ** | −4.28 |
| >500 km | 4.32 | 9.32 * | −5.23 |
| >550 km | 5.97 | 6.45 | 4.11 |
| >600 km | 2.91 | 7.35 | 2.67 |
| >700 km | 2.01 | 4.45 | 1.36 |
| >800 km | 0.97 | 1.87 | −0.23 |
| >900 km | 0.35 | 1.11 | 0.15 |
| >1000 km | 0.86 | 0.76 | 0.11 |

Note: Standard errors are clustered at the provincial level; *** $p < 0.01$, ** $p < 0.05$, * $p < 0.1$.

*3.6. Threshold Effect of City Size*

City innovation is based on the agglomeration and diffusion of different innovative elements, particularly the circulation of innovative talent across regions and departments, which facilitates the transmission and diffusion of knowledge, thus driving the dynamic growth of the innovation system [21]. Generally, enterprise productivity and talent structure vary significantly between different city sizes [11]. In bigger cities, the more high-tech enterprises and innovative talents there are, the more capital and technology. Thus, it is easier to take advantage of the opportunities provided by the HSR network and propel cities towards innovation-driven growth. In large cities, the level of economic activity is much higher, leading to an increase in energy consumption and pollution. This has resulted in a greater demand for green innovation that takes into account environmental performance. On the other hand, smaller cities usually face a disadvantage environment when it comes to innovation factor aggregation, as they have a weaker innovation base, making it difficult to achieve economies of scale. Furthermore, they cannot benefit from the advantages of talent, industry, and capital aggregation facilitated by the construction of HSR networks, and the new knowledge generated in other regions is often structurally different from local existing knowledge due to the difference of industry structure, which cannot be easily converted into economic growth like in large cities. Although different cities are connected in the HSR network, the basis for the aggregation of innovative elements differs between cities, and this results in a significant sorting effect based on the threshold of the city's size.

The results of Table 8 show that both the traditional OLS and SDM estimations have a significantly positive estimation coefficient for the interaction between population size and HSR network when it comes to city innovation performance. This implies that cities with larger populations are more likely to benefit from the opportunities provided by the HSR network, thus promoting their innovation. Moreover, the spatial estimation coefficient of the interaction term between population size and HSR network is also significantly positive, indicating that city size not only has a significant threshold effect on city innovation but also strengthens the threshold effect of neighboring cities on innovative development. In short, city size strengthens the innovative developments of HSR networks to larger cities through the spatial sorting of innovative elements.

**Table 8.** Threshold effect of city size on HSR network effect.

| | OLS | SDM |
|---|---|---|
| Degree | 128.812 * | −17.985 |
| | (73.049) | (11.529) |
| Population | 0.015 * | 0.016 ** |
| | (0.008) | (0.007) |
| Degree*Population | 0.201 * | 0.073 ** |
| | (0.101) | (0.031) |
| $\theta$ (W*Degree) | | 41.010 ** |
| | | (17.164) |
| $\theta$ (W*Degree*Population) | | 0.103 *** |
| | | (0.036) |
| $\rho$ (Innovation Index) | | 0.362 *** |
| | | (0.089) |
| Control variable | ✓ | ✓ |
| Fixed time | ✓ | ✓ |
| Fixed city | ✓ | ✓ |
| Obs | 2508 | 2508 |
| $R^2$ | 0.586 | 0.083 |

Note: Standard errors are reported in the parentheses and clustered at the provincial level; *** $p < 0.01$, ** $p < 0.05$, * $p < 0.1$.

## 4. Mechanism Analysis

HSR networks are a crucial factor in innovation, as they facilitate the aggregation and circulation of innovative elements. This transportation system links cities, reduces the distance between regions, and encourages face-to-face communication. This, in turn, leads to industrial concentration and diffusion, which attracts investment and encourages the innovative developments of cities. This process of knowledge spillover between cities ultimately results in a pattern of strong and weak cities in terms of innovation capacity.

### 4.1. Talent Agglomeration and Mobility

According to the endogenous growth theory, regional innovation growth is largely dependent on R&D investment and knowledge spillover. R&D investment includes talents and capital, while the agglomeration and mobility of talents is the most important factor in the transmission of knowledge, especially for the implicit knowledge shared among cities. The availability of HSRs can facilitate the mobility between regions and talent aggregation, which in turn facilitates knowledge spillover between cities. On the one hand, talent is the main source of knowledge creation, as it can foster knowledge sharing and technological cooperation among the components of innovation. Moreover, HSR networks facilitate the agglomeration of talents in cities, which bring about scale effects through labor pool effects and knowledge spillovers, thus enhancing city innovation. On the other hand, the construction of HSR networks facilitates the movement of innovative talents between cities, leading to technology spillovers. This wider diffusion of knowledge and technology allows for an increased exchange of ideas between cities, resulting in an improved level of technological innovation. In other words, the extension of HSR networks enhances the sharing and exchange of technology spillovers, thereby boosting the overall technological innovation. The increased connectivity of HSR networks allows for the free mobility of innovative talent between cities, enabling individuals who are disadvantaged in terms of knowledge and technology to acquire the expertise of those with comparative advantages. This also leads to a technology spillover effect, which will not only encourage imitation among innovative talents but will also help to raise the innovation levels in neighboring cities and increase the correlation between city innovation. The theory of knowledge absorption capacity notes that the level of human capital agglomeration in an area must exceed a certain "threshold" to effectively promote innovation [22]. That is, the size of city

plays a role in the agglomeration of innovative talents, and larger cities are more likely to take advantage of the advantages of HSR networks to promote innovative developments.

This article evaluates the influence of the HSR networks on city innovation through the analysis of urban employments and passenger volumes. It also examines the threshold effect of talent aggregation moderated by city size, which aims to answer the question of which cities can benefit from the construction of an HSR network and where the capital that these cities gain comes from. The results from columns (1) to (3) of Table 9 suggest that the HSR network not only expands local employment, but also intensifies the agglomeration effect of nearby employment. Thus, the HSR network encourages the innovative growth of cities and its spatial spillover effect through the agglomeration of talents. HSR networks not only facilitate the agglomeration of talents in cities but also boost the knowledge diffusion and innovative growth of cities by hastening the mobility and communication of talents between cities. The results of population mobility effects estimated from the intercity passenger volume in columns (4) to (6) of Table 9 support the hypothesis that HSR network connections not only increase the scale of local population mobility but also intensify the scale of the neighboring population's mobility through the HSR network index. Thus, HSR networks facilitate the innovative developments and spatial spillover effects of cities through the mobility and communication of talents. Columns (3) and (6) of Table 9 show that the influence of HSR networks on city talent aggregation and mobility is amplified in larger cities. This suggests that there is a threshold of city size which has an effect on talent aggregation and mobility. In short, HSR networks not only promote the spatial spillover of knowledge through the agglomeration and mobility of talents but also realize the innovative developments of cities by absorbing the human capital of surrounding areas, especially the human capital around large-scale cities.

**Table 9.** The impact of HSR networks on employment and passenger traffic.

| | Urban Employment | | | Passenger Volume | | |
|---|---|---|---|---|---|---|
| | OLS | SDM | SDM | OLS | SDM | SDM |
| Degree | 170.5 *** | 35.89 ** | −27.19 | 5.096 ** | 0.256 * | 0.573 ** |
| | (11.6) | (15.60) | (47.52) | (2.295) | (0.156) | (0.027) |
| Population | 0.035 *** | 0.109 * | 0.099 * | 0.001 *** | −0.001 | −0.003 |
| | (0.010) | (0.065) | (0.049) | (0.001) | (0.003) | (0.013) |
| Degree*Population | 0.305 ** | | 0.153 *** | 0.008 *** | | 0.011 ** |
| | (0.143) | | (0.010) | (0.003) | | (0.005) |
| $\theta$ (W*Degree) | | 42.02 * | 115.6 | | 2.883 *** | −0.687 |
| | | (23.72) | (180.9) | | (0.928) | (5.731) |
| $\theta$ (W*Degree*Population) | | | 0.355 *** | | | 0.108 ** |
| | | | (0.026) | | | (0.040) |
| Control variable | √ | √ | √ | √ | √ | √ |
| Fixed time | √ | √ | √ | √ | √ | √ |
| Fixed city | √ | √ | √ | √ | √ | √ |
| Obs | 2508 | 2508 | 2508 | 2508 | 2508 | 2508 |
| $R^2$ | 0.474 | 0.083 | 0.065 | 0.568 | 0.053 | 0.076 |

Note: Standard errors are reported in the parentheses and clustered at the provincial level; *** $p < 0.01$, ** $p < 0.05$, * $p < 0.1$.

### 4.2. Financial Capital Agglomeration

According to the endogenous growth theory, regional innovation growth is not only derived from knowledge spillovers but also from R&D investments. HSR networks can improve the agglomeration effect of financial capital, thus significantly promoting city innovation. By providing HSR connections between cities, accessibility can be improved, which declines the cost, time, and effort spent by credit supply and demand parties in negotiations, communication, and cooperation. This will consequently decrease the time for credit supply and demand parties to find suitable partners and speed up the matching

of transaction parties. On the other hand, HSR connections expand the concept of distance between cities from the original "geographical distance" to "time distance", which means that financial institutions can more easily access distant investment projects, thus reducing the cost of information search and helping to alleviate the problem of "adverse selection" before credit investments and the "moral hazard" after an investment [23]. HSR networks have eliminated geographical restrictions on the movement of financial resources, allowing them to concentrate in cities with HSR connections. This has helped to reduce the financing restriction of firms and encouraged them to innovate. However, the mobility of financial resources is guided by the "rational economist" hypothesis, meaning that resources will be directed towards cities that offer a greater economic growth potential and higher returns, thus creating a threshold effect on the agglomeration of financial capital.

This paper uses foreign investments and the balance of loans to explore the driving mechanism that HSR networks use to affect city innovation, as well as the threshold effect of financial capital agglomeration through the regulatory role of city size. Table 10 reveals that the HSR network not only boosts the foreign investment and balance of loans in local cities but also amplifies the agglomeration effect of foreign investment and loan balances in nearby cities. Consequently, HSR networks facilitate the innovation of cities and their spatial spillover effects through the agglomeration of financial capital. Table 10 reveals that city size has a threshold effect on financial capital agglomeration when it comes to the influence of HSR networks. It appears that HSR networks can help to alleviate the financing constraints of city innovation by accumulating financial capital, and this effect is amplified in larger cities. In this way, cities have enough capital to invest, thus enabling the city to achieve innovative developments.

**Table 10.** The impact of HSR networks on foreign investments and loan balances.

| | Actual Foreign Investment | | | Balance of Loans from Financial Institutions at Year End | | |
|---|---|---|---|---|---|---|
| | OLS | SDM | SDM | OLS | SDM | SDM |
| Degree | 20.901 *** | 5.276 *** | 7.212 | 14.766 ** | 8.565 * | 4.673 ** |
| | (7.365) | (1.604) | (7.675) | (6.567) | (0.793) | (2.027) |
| Population | −0.007 *** | 0.003 | 0.009 * | 0.413 | −0.238 | 0.347 *** |
| | (0.002) | (0.005) | (0.005) | (0.367) | (0.243) | (0.143) |
| Degree*Population | 0.072 | | 0.047 *** | 0.256 ** | | 1.348 * |
| | (0.068) | | (0.012) | (0.121) | | (0.727) |
| $\theta$ (W*Degree) | | 12.216 ** | 16.342 * | | 5.286 ** | 7.346 |
| | | (5.729) | (10.116) | | (2.845) | (4.967) |
| $\theta$ (W*Degree*Population) | | | 0.478 ** | | | 2.567 *** |
| | | | (0.221) | | | (0.923) |
| Control variable | √ | √ | √ | √ | √ | √ |
| Fixed time | √ | √ | √ | √ | √ | √ |
| Fixed city | √ | √ | √ | √ | √ | √ |
| Obs | 2058 | 2508 | 2058 | 2508 | 2058 | 2508 |
| $R^2$ | 0.614 | 0.083 | 0.067 | 0.473 | 0.058 | 0.081 |

Note: Standard errors are reported in the parentheses and clustered at the provincial level; *** $p < 0.01$, ** $p < 0.05$, * $p < 0.1$.

### 4.3. Industrial Agglomeration

HSRs have facilitated economic agglomeration, but does economic agglomeration affect the spillover effect of technological innovation? The innovation theory holds that "innovation is not an isolated behavior of enterprises, it can be realized by the agglomeration of enterprises" [24]. Furthermore, new products developed in an industry often have an indirect impact on the production technologies of other enterprises or industries by putting them into the industrial chain [25]. The aggregation of industries in a special area can foster a sense of mutual understanding and trust between enterprises, which in turn facilitate the communication and sharing of innovative ideas, thus leading to the spread of innovation within the industry and the formation of a learning environment [26,27]. The aggregation of industries has resulted in lower transaction costs, larger market sizes, and increased

labor agglomeration, which have a motivating effect on market growth [28]. In conclusion, HSR networks boost city innovation and knowledge spillovers by enhancing the level of industrial agglomeration. Similarly, the size of industrial agglomeration still has a threshold effect on city innovation. Cities with higher levels of industrial agglomeration can provide more abundant innovative ideas for innovation entities within the city, creating a more favorable innovation atmosphere for the development of innovation entities, and thus the scale effect of the city innovation factor agglomeration is more easily seen.

This article employs the variables of total output and number of enterprises above designated size to examine the driving mechanism of industrial agglomeration for HSR networks that affect city innovation and further examines the threshold effect of industrial agglomeration through the regulatory effect of city size. Table 11 demonstrates that the HSR network connection not only boosts the total output and number of enterprises above designated size in a local city but also intensifies the agglomeration effect of the total output and number of enterprises above designated size in other nearby cities. Thus, HSR networks promote the innovative growth and spatial spillover effect of cities through industrial agglomeration. Table 11 also shows that city size strengthens the impact of HSR networks on industrial agglomeration, indicating a threshold effect of city size on industrial agglomeration. Consequently, the industrial agglomeration effect of HSR networks on larger cities is more pronounced. All in all, HSR networks promote the spatial spillover effect of knowledge through industrial agglomeration, and the city size further boosts this promoting effect, thereby achieving innovative developments of the city.

**Table 11.** The influence of an HSR network on enterprise above designated size.

|  | Total Output of Enterprise above Designated Size | | | Number of Enterprises above Designated Size | | |
|---|---|---|---|---|---|---|
|  | OLS | SDM | SDM | OLS | SDM | SDM |
| Degree | 35.762 *** | 24.349 | 11.456 | 4.234 *** | 2.568 | 2.567 * |
|  | (9.568) | (15.953) | (19.346) | (1.789) | (4.834) | (1.359) |
| Population | 0.027 ** | 0.456 *** | 0.417 | 1.349 | −1.523 | −0.518 * |
|  | (0.013) | (0.141) | (0.345) | (0.923) | (1.036) | (0.248) |
| Degree*Population | 7.346 ** |  | 4.267 * | 1.458 * |  | 0.561 |
|  | (3.568) |  | (2.312) | (0.821) |  | (1.937) |
| θ (W*Degree) |  | 3.549 ** | 7.234 ** |  | 0.346 *** | 6.289 *** |
|  |  | (1.729) | (3.267) |  | (0.105) | (1.593) |
| θ (W*Degree*Population) |  |  | 2.578 ** |  |  | 1.001 *** |
|  |  |  | (1.249) |  |  | (0.367) |
| Control variables | √ | √ | √ | √ | √ | √ |
| Fixed year | √ | √ | √ | √ | √ | √ |
| Fixed city | √ | √ | √ | √ | √ | √ |
| Obs | 2058 | 2508 | 2058 | 2508 | 2058 | 2058 |
| $R^2$ | 0.734 | 0.045 | 0.049 | 0.598 | 0.051 | 0.056 |

Note: Standard errors are reported in the parentheses and clustered at the provincial level; *** $p < 0.01$, ** $p < 0.05$, * $p < 0.1$.

## 5. Conclusions

HSR networks cause a time–space compression effect which has a great influence on the mobility of innovation factors and the spatial transformation of innovation capability. Therefore, how to boost the mobility of innovation factors and improve the spatial allocation of innovation resources through HSRs and other transport infrastructures is important to city innovation. Based on the perspective of the geographical heterogeneity of 'New' new economic geography, this paper adopts the panel data of Chinese cities from 2008 to 2019 and the method of CSDIDs to study the impact of HSR networks on city innovation. The findings of this study demonstrate that cities have a notable spatial autocorrelation of innovation. The convenience of travelling between cities, changed by the HSR network, has been continuously stimulating the enhancement of city innovative capabilities, as well as fostering the innovative capability of neighboring cities, indicating that China's HSR construction has a strong spatial effect on city innovation. However, it was found that

the spatial spillover effect of the opening of HSRs on city innovation has a geographical attenuation characteristic. The effective scope of the spatial spillover effect of HSRs on city innovation is within the radius of 400 km, and this effect is no longer significant when the distance exceeds the distance of 400 km. HSR networks have a significant impact on city innovation, mainly through the promotion of knowledge spillover by the agglomeration of talent, financial capital, and industry. Furthermore, this innovative effect is regulated by the city's size, as larger cities are more capable of obtaining the dividend effect of talent agglomeration, financial capital agglomeration, and industrial agglomeration from the surrounding areas, so as to achieve a new spatial pattern of innovative developments in these cities that is covered by HSRs.

When investing in large-scale transportation infrastructures, China should pay attention to its role on the spatial reconstruction of the city's innovation capacity. The government should give more attention to the local and neighboring promoting innovation effects of the HSR network and gradually form a balanced development trend of regional innovation. In addition, as the core of institutional innovation, the government needs to coordinate the relationship between various innovation units and strengthen the combination of production, university, and research, thus improving the overall level of city innovation. Finally, smaller cities should make full use of the spatial spillover effect brought by HSR networks, actively undertake the industries and labor transferred from other cities, and strive to offset the siphon effect through the spillover effect of larger cities, thus realizing the innovative developments of smaller cities.

**Author Contributions:** Conceptualization, K.Z.; Methodology, K.Z.; Validation, W.L.; Formal analysis, K.Z.; Investigation, W.L.; Resources, K.Z.; Data curation, K.Z.; Writing—original draft, K.Z.; Writing—review & editing, W.L.; Project administration, W.L. All authors have read and agreed to the published version of the manuscript.

**Funding:** This research received no external funding.

**Institutional Review Board Statement:** Not applicable.

**Informed Consent Statement:** Not applicable.

**Data Availability Statement:** The data presented in this study are available on request from the corresponding author.

**Conflicts of Interest:** The authors declare no conflict of interest.

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
