# Peer review of "High-Speed Railway Network, City Heterogeneity, and City Innovation"

_sustainability, doi:10.3390/su152115648_

Round 1

Reviewer 1 Report

Comments and Suggestions for Authors

1.     The research value of the article is reflected in achieving a balanced spatial pattern of urban innovation, which has strong practical significance. However, the paper does not clearly explain the innovation of the research. Readers and reviewers would like to know what theoretical contributions have been made in this field of study? What is the unique academic value reflected? This is also very important.

2.     In the research review, the number of references is relatively small, and there are very few references covering the years 2021/2022/2023. The timeliness of the references is not strong, and the most important thing is that there are few high-quality literature, which neither demonstrates a solid disciplinary foundation nor comments on the mainstream and forward-looking trends of current research. It is recommended that the author carefully supplement and improve.

3.     One of the core keywords of the article is spatial heterogeneity, but there is no spatial map in the entire text, let alone research methods and related conclusion analysis of spatial geography. How is spatial heterogeneity achieved? Where is it high? Where is it low? What are typical areas? None of these are clear.

4.      The conclusion section of the article is too short, and it is suggested to add a policy suggestion section to supplement the corresponding more targeted and specific countermeasures.

Author Response

Question #1: The research value of the article is reflected in achieving a balanced spatial pattern of urban innovation, which has strong practical significance. However, the paper does not clearly explain the innovation of the research. Readers and reviewers would like to know what theoretical contributions have been made in this field of study? What is the unique academic value reflected? This is also very important.

Author Reply: Really thank you for your recommendation. We have technically edited the introduction and contribution, which is easily understand what we want to research and what we can do to contribute the existed literature, so that it can easily follow the manuscript, shown as following:

The marginal contributions of this paper are as follows: First, traditional studies on the impact of HSR on city innovation based on dummy variable ignores the dynamic change process of HSR network expansion. This paper identifies the impact of HSR network on city innovation and its spatial spillover effect through CSDID method, based on HSR network index instead of dummy variable of whether the city has been covered by HSR; Second, most of the existing studies studied the impact of HSR on city innovation under the assumption of homogeneity, they ignore the city heterogeneity regarding the difference of receiving and transforming knowledge under different city size. This paper examines heterogeneity effect of HSR network on city innovation under the moderating effect of city size, which demonstrates the agglomeration and diffusion difference between different city size. Third, previous studies have demonstrated the mechanism of city innovation by factor agglomeration and knowledge spillover, but it still do not know the path that how do these innovative elements agglomeration and diffusion in the HSR network, and what is the criteria for determining the direction of factor agglomeration and diffusion, this paper adopts CSDID to explore its spatial mechanism and the agglomeration and diffusion path of innovation elements, providing a theoretical basis for spatial allocation of innovation resource under different city size.

Question #2: In the research review, the number of references is relatively small, and there are very few references covering the years 2021/2022/2023. The timeliness of the references is not strong, and the most important thing is that there are few high-quality literature, which neither demonstrates a solid disciplinary foundation nor comments on the mainstream and forward-looking trends of current research. It is recommended that the author carefully supplement and improve.

Zeng, Zhixin, Long Zhao, and Xiaojun Wang. "Does improved transportation promote innovation? Evidence from China’s cities." Applied economics 54.23 (2022): 2643-2657.

Fan, Xiaomin, and Yingzhi Xu. "Does high-speed railway promote urban innovation? Evidence from China." Socio-Economic Planning Sciences 86 (2023): 101464.

Hou, Yuting. Agglomeration spillover, accessibility by high-speed rail, and urban innovation in China: A focus on the electronic information industry. Habitat International 126 (2022): 102618.

Author Reply: Really thank you for your recommendation for the literature. We have replaced and updated these literatures by more high-quality and latest literature, which are more relevant to our article, shown in the full article.

Question #3:  One of the core keywords of the article is spatial heterogeneity, but there is no spatial map in the entire text, let alone research methods and related conclusion analysis of spatial geography. How is spatial heterogeneity achieved? Where is it high? Where is it low? What are typical areas? None of these are clear.

Author Reply: Really thank you for your recommendation. We have revised the article and added the Moran scatter plot for innovation index and authorized patent and four maps for HSR network index regarding different centrality, which are shown in the variable construction section. In addition, we mainly use the spatial model of SDM to analysis the problem of spatial geography. In our article, the crucial heterogeneity is city size, which are divided into two groups by large city and small city based on their GDP ranking in the base period, so we mainly use the city size to analysis the heterogeneity and its spatial heterogeneity by OLS and SDM, which is introduced into the basic model by the interaction term of HSR and city size. Our results show that the large city strengthen the promoting effect of HSR on city innovation.

Question #4:  The conclusion section of the article is too short, and it is suggested to add a policy suggestion section to supplement the corresponding more targeted and specific countermeasures.

Author Reply: Really thank you for your recommendation. We have added a policy suggestion section based on our analysis results, which is corresponding more targeted and specific countermeasures, shown as following:

When investing in large-scale transportation infrastructure, China should pay attention to its role on spatial reconstruction of city innovation capacity. The government should give more attention to the local and neighboring promoting innovation effect of the HSR network, and gradually form a balanced development trend of regional innovation. In addition, as the core of institutional innovation, the government needs to coordinate the relationship between various innovations units and strengthen the combination of production, university and research, thus improving the overall level of city innovation. Finally, smaller cities should make full use of the spatial spillover effect brought by HSR network, actively undertake the industries and labor transferred from other cities, and strive to offset the siphon effect through the spillover effect of larger cities, thus realizing the innovative development of smaller cities.

Reviewer 2 Report

Comments and Suggestions for Authors

This issue is an interesting and ambitious research. The authors attempted to study the the HSR network and its effect on city innovation from the perspective of “New’ new economic geography” with 2008-2019 panel data (S-DID). The proposed model and analysis are good, but the results are not interesting (some seem to be inconsistent with existing research, for example, the geographical boundary of 400km ) and make the readers confused that what this research would tell us and what the innovation of this manuscript?

The authors argument research has not adequately addressed the spatial disparities of the HSR network and their effects on city innovative maybe need to be discussed. As I know, in terms of city size (mega cities, medium-sized cities, small cities), city/urban innovation and the HSR, spatial/ knowledge spillover, agglomeration and the HSR, optimal distance, accessibility and the HSR, many existing research(as follows) focus on the effects of HSR on innovation.

I feel sorry that I reject this paper, the author may need to find what research questions deduced from the niches of literature review about the HSR network and urban innovation in China.

Zeng, Zhixin, Long Zhao, and Xiaojun Wang. "Does improved transportation promote innovation? Evidence from China’s cities." Applied economics 54.23 (2022): 2643-2657.

Fan, Xiaomin, and Yingzhi Xu. "Does high-speed railway promote urban innovation? Evidence from China." Socio-Economic Planning Sciences 86 (2023): 101464.

Hou, Yuting. Agglomeration spillover, accessibility by high-speed rail, and urban innovation in China: A focus on the electronic information industry. Habitat International 126 (2022): 102618.

Comments on the Quality of English Language

As above comments

Author Response

Question #1: This issue is an interesting and ambitious research. The authors attempted to study the HSR network and its effect on city innovation from the perspective of “New’ new economic geography” with 2008-2019 panel data (S-DID). The proposed model and analysis are good, but the results are not interesting (some seem to be inconsistent with existing research, for example, the geographical boundary of 400km ) and make the readers confused that what this research would tell us and what the innovation of this manuscript?

Author Reply: Really thank you for your recommendation. I am very sorry for the confused sentence, this sentence mainly wants to show the attenuation effect for the spatial spillover of city innovation with the increase of geographical distance, which have been revised by “the innovation accelerative effect gradually decreases as the distance from the city covered by HSR increases, and completely disappears at the distance of 400km.” In addition, we also revised the introduction and contribution, which is easily understand what we want to research and what we can do to contribute the existed literature, we think these revises can strengthen our innovation and researching importance.

Question #2: The authors’ argument “research has not adequately addressed the spatial disparities of the HSR network and their effects on city innovative” maybe need to be discussed. As I know, in terms of “city size (mega cities, medium-sized cities, small cities), city/urban innovation and the HSR”, “spatial/ knowledge spillover, agglomeration and the HSR”, “optimal distance, accessibility and the HSR”, many existing research(as follows) focus on the effects of HSR on innovation.

Author Reply: Really thank you for your recommendation. In our article, the crucial innovation is introducing the heterogeneity of city size from the spatial perspective, so we mainly use the city size to analysis the heterogeneity and its spatial heterogeneity by OLS and SDM, which is introduced into the basic model by the interaction term of HSR and city size. Our results show that the large city strengthen the promoting effect of HSR on city innovation on both local cities and neighboring cities. In addition, we have revised the introduction and contribution, which is easily understand what we want to research and what we can do to contribute the existed literature, we think these revises can strengthen our innovation and researching importance, shown as following:

The marginal contributions of this paper are as follows: First, traditional studies on the impact of HSR on city innovation based on dummy variable ignores the dynamic change process of HSR network expansion. This paper identifies the impact of HSR network on city innovation and its spatial spillover effect through CSDID method, based on HSR network index instead of dummy variable of whether the city has been covered by HSR; Second, most of the existing studies studied the impact of HSR on city innovation under the assumption of homogeneity, they ignore the city heterogeneity regarding the difference of receiving and transforming knowledge under different city size. This paper examines heterogeneity effect of HSR network on city innovation under the moderating effect of city size, which demonstrates the agglomeration and diffusion difference between different city size. Third, previous studies have demonstrated the mechanism of city innovation by factor agglomeration and knowledge spillover, but it still do not know the path that how do these innovative elements agglomeration and diffusion in the HSR network, and what is the criteria for determining the direction of factor agglomeration and diffusion, this paper adopts CSDID to explore its spatial mechanism and the agglomeration and diffusion path of innovation elements, providing a theoretical basis for spatial allocation of innovation resource under different city size.

Question #3: I feel sorry that I reject this paper, the author may need to find what research questions deduced from the niches of literature review about the HSR network and urban innovation in China.

Author Reply: Really thank you for your recommendation. We are really sorry that we have not exactly provided the research questions and innovation, while we have revised the full article and strengthen the research question and innovation in the introduction section, we think these revises can strengthen our innovation and researching importance, shown as following:

In 2016, China introduced the “National Innovation Driven Development Strategy Outline”, making innovation as a popular subject in Chinese economic research. Economically speaking, the combination of factors such as talent, capital, and technology is a major component in fostering innovation. According to the endogenous growth theory, regional innovation growth is mainly due to R&D investment and knowledge spillovers (Romer, 1990). However, the mobility of resources, labor, and capital is restricted by distance, making it difficult to form an appropriate spatial distribution structure. Optimizing the spatial arrangement of innovative resources and increasing city innovation efficiency is essential for building an innovative nation and achieving innovation-driven development in China.

HSR has been identified as one of the most revolutionary and innovative forms of transportation in the 21st century. It has been shown to reduce the costs of knowledge spillover by shortening spatial and temporal distances, as well as accelerating the flow of innovative elements such as talent and capital between cities (Wang and Cai, 2020; Wang et al., 2022). Additionally, it has been found to influence the spatial distribution of economic factors, thus reshaping the regional spatial development structure. Furthermore, HSR has been observed to reduce the cost of trade between cities (Wang et al., 2018; Fingleton and Szumilo, 2019). The New Economic Geography theory suggests that when trade costs are reduced, production factors will be relocated in order to increase actual profits, and the relocation of innovative elements across regions will cause a spatial differentiation of city creativity (Krugman, 1991). 

HSR can be beneficial for economic growth and the aggregation of economic activities. The network’s characteristics enable the transfer of production factors from one area to another, and cities that are close to each other are more likely to take advantage of the resources of the central city, resulting in a positive spatial spillover effect (Dong, 2020; Gao and Zheng, 2020). However, in regions with a well-developed HSR network, the economic agglomeration effect of the central city can also impede the economic development of the surrounding cities, resulting in a negative spatial spillover effect (Boarnet, 1998). This can be detrimental to poorer areas, as they cannot benefit from the increased investment in nearby areas (Lvarez et al., 2016). The rationale explanation for coming to a conflicting conclusion is that various cities have diversity at different stages, and blurring different spatial patterns will cause a spatial mismatch in the innovative development of cities. Exploring the innovative difference and its mechanism between different city size is the most urgent problem that needs to be solved for city innovation from the global perspective.

This article proposes that in order to reduce uncertainty in the innovative development of cities, mapping the stages of city development to changes in the spatial structure should be done, which can reveal the relationship between the dynamic changes caused by the construction of the HSR network and the innovation output of different cities. Thus, it is necessary to consider how the implementation of the HSR network changes the spatial pattern of city innovation and boosts the spatial allocation of innovative resources between cities. Ignoring this would hinder the efficient allocation of innovative resources by the government and enterprises, as well as the optimization and balanced growth of innovation patterns between cities.

Industrial transfer is typically arranged geographically based on the cost-minimizing principle, meaning even smaller sized cities can take part in specific industrial transfers due to their lower production and living costs (Coe and Helpman, 1995). However, innovation activities have a higher skill threshold, and the time cost for talents is more significant, so the skill threshold due to different city size can be a large limitation on innovation (Storper and Scott, 2009; Zhang, 2020). Numerous facts point to the fact that global innovation resources are increasingly concentrated in cities with a great deal of talent and technological capital. These cities not only demonstrate individual innovation strength, but also demonstrate collective innovation abilities (Gao Xiang, 2015; Wang et al., 2021). However, in cities that all have been covered by HSR, smaller cities usually cannot take advantage of the innovation opportunity that HSR brings, what accounts for these disparities between different city size.

In the last few years, research on city agglomeration has revealed that the notion of homogeneity among enterprises is flawed, and that disparities in enterprises efficiency have a major effect on city innovative efficiency. Melitz and Ottaviano (2008), Ottaviano (2011), and Combes et al. (2012) have proposed more realistic hypotheses in this regard. Agglomeration economies can certainly increase enterprises productivity, but the ‘New’ new economic geography theory also indicates that the high cost and competitive environment of large cities can act as a filter for enterprise production efficiency. As the market size grows, the competition among firms intensifies, leading to a decrease in the critical marginal production cost that determines which firms can survive. Firms that fail to reach this cost threshold will not be able to survive in this market. These explained the reason that global innovation resources are increasingly concentrated in large cities, but it still does not clearly know the mechanism that HSR promotes the agglomeration and diffusion between different innovation elements, there are few studies that focusing on heterogeneity city innovation from the spatial perspective, traditional viewpoint of knowledge spillover and agglomeration does not consider the city heterogeneity and spatial perspective simultaneously, so it does not exactly capture the direction of factor agglomeration and diffusion by using traditional analysis method and perspective. However, heterogeneity is the core of city innovation for determining the direction of factor agglomeration and diffusion in real world.

This investigation has established a robust theoretical basis regarding the spillover effect of transportation infrastructure and city innovation. Nevertheless, most of the current studies are based on the homogeneity hypothesis of the new economic geography theory, and neglect the threshold effect of different city size on the selection of innovative elements. The spatio-temporal compression effect of HSR network significantly affects the circulation of innovative elements and the spatial arrangement of innovation capacity, which have yet to be fully understood. This article examines the impact of HSR networks on city innovation by utilizing the continuous spatial difference in differences (CSDID) method. This method is an improvement on the DID, which only use the dummy variable by whether HSR is open or not in previous studies. The CSDID method takes into account the annual construction process of HSR networks and the gradual state between cities, allowing for the dynamic process of HSR network expansion to be taken into account. This method also establishes a connection between the connectivity of city networks in different spatial structure changes and innovation output, thus providing a better understanding of the structural effects of HSR networks on city innovation.

The marginal contributions of this paper are as follows: First, traditional studies on the impact of HSR on city innovation based on dummy variable ignores the dynamic change process of HSR network expansion. This paper identifies the impact of HSR network on city innovation and its spatial spillover effect through CSDID method, based on HSR network index instead of dummy variable of whether the city has been covered by HSR; Second, most of the existing studies studied the impact of HSR on city innovation under the assumption of homogeneity, they ignore the city heterogeneity regarding the difference of receiving and transforming knowledge under different city size. This paper examines heterogeneity effect of HSR network on city innovation under the moderating effect of city size, which demonstrates the agglomeration and diffusion difference between different city size. Third, previous studies have demonstrated the mechanism of city innovation by factor agglomeration and knowledge spillover, but it still do not know the path that how do these innovative elements agglomeration and diffusion in the HSR network, and what is the criteria for determining the direction of factor agglomeration and diffusion, this paper adopts CSDID to explore its spatial mechanism and the agglomeration and diffusion path of innovation elements, providing a theoretical basis for spatial allocation of innovation resource under different city size.

Reviewer 3 Report

Comments and Suggestions for Authors

Congratulations for the article. Some minor aspects can be improve:

- figure 1 is with not reference in text

- reference about Table 2 is not located before table

- about conclusions: note you that innovation process depend of many variables and HSR services is only one of possibles variables that can help to innovation in determinate conditions. It is possible innovation without patents that is the only variable used to stablish statistical correlations.

Comments on the Quality of English Language

Please, to review minor grammar mistakes (lines 179, 279, 325, 282 ...)

Author Response

Congratulations for the article. Some minor aspects can be improve:

Author Reply: Really thank you for your valuable comments. We are pleased to hear the comments for our article. All the explanations regarding the revisions of our manuscript are as follows with point-by-point response.

Question #1: figure 1 is with not reference in text

Author Reply: Really thank you for your valuable comments. We have added the reference in the text.

Question #2: reference about Table 2 is not located before table

Author Reply: Really thank you for your valuable comments. We have added the reference in the text.

Question #3: about conclusions: note you that innovation process depend of many variables and HSR services is only one of possibles variables that can help to innovation in determinate conditions. It is possible innovation without patents that is the only variable used to stablish statistical correlations.

Author Reply: Really thank you for your valuable comments. This paper mainly studies the impact of HSR network on city innovation, so we mainly focus on HSR network and its spatial effect, we will deepen the analysis framework of the impact on city innovation in the future. In addition, it is indeed that patents are only one form of city innovation, and innovation should be a comprehensive reflection of various capabilities. Therefore, we use multiple dimensions such as patent index and authorized patents to present city innovation. Of course, there are other indicators to reflect the level of city innovation, and we will further deepen the expression of innovation ability in the follow-up research.

Question #4: Please, to review minor grammar mistakes (lines 179, 279, 325, 282 ...)

Author Reply: Really thank you for your valuable comments. We have revised the grammar mistakes.

Round 2

Reviewer 1 Report

Comments and Suggestions for Authors

1.      The author team basically answered my questions comprehensively. After revising and improving some minor doubts and suggestions, I agree to recommend publication.

2.      In terms of academic value and significance, it is recommended to add theoretical contributions from urban geography and transportation geography; The role and value of this manuscript in China's transportation power and other strategies. The writing ideas for the following two references are recommended for learning.

[1] Zhao Pengjun, Lyu Di, Hu Haoyu, et al. The people-oriented land passenger transportation regionalization in China. Acta Geographica Sinica, 2023,78(6):1498-1514.

[2] Tian, Shenzhen., Jiang, Jialin., Li, Hang. et al. Flow space reveals the urban network structure and development mode of cities in Liaoning, China. Humanities and Social Sciences Communications  10, 257 (2023). https://doi.org/10.1057/s41599-023-01752-5.

Author Response

Question #1: The author team basically answered my questions comprehensively. After revising and improving some minor doubts and suggestions, I agree to recommend publication.

Author Reply: Really thank you for your recommendation.

Question #2:   In terms of academic value and significance, it is recommended to add theoretical contributions from urban geography and transportation geography; The role and value of this manuscript in China's transportation power and other strategies. The writing ideas for the following two references are recommended for learning.

[1] Zhao Pengjun, Lyu Di, Hu Haoyu, et al. The people-oriented land passenger transportation regionalization in China. Acta Geographica Sinica, 2023,78(6):1498-1514.

[2] Tian, Shenzhen., Jiang, Jialin., Li, Hang. et al. Flow space reveals the urban network structure and development mode of cities in Liaoning, China. Humanities and Social Sciences Communications  10, 257 (2023). https://doi.org/10.1057/s41599-023-01752-5.

Author Reply: Really thank you for your recommendation. we have revised the contribution, which is easily understand what we want to research and what we can do to contribute the existed literature, we think these revises can strengthen our theoretical innovation and researching importance, shown as following:

Finally, this study further deep the theory geographic economics by proposing the select effect of city size when HSR network facilitate the element flow, which dose not only focuses on the agglomeration and diffusion effect of absolutely large and small city, it should carry out in-depth promotion in combination with China’s “Metropolitan Circle” form an efficient and comprehensive network structure of city innovation based on the ordered city size and city’s functions and roles.

Reviewer 2 Report

Comments and Suggestions for Authors

Thank you for the authors' efforts, I have no other suggestions for the revised version.

Author Response

Question #1: Thank you for the authors' efforts, I have no other suggestions for the revised version.

Author Reply: Really thank you for your recommendation.
